# OBJLOC: INDOOR CAMERA RELOCALIZATION BASED ON OPEN-VOCABULARY OBJECT-LEVEL MAPPING

## ABSTRACT

Indoor visual relocalization plays a key role in emerging spatial and embodied AI applications. However, prior research has predominantly focused on methods based on low-level vision. Despite notable progress, these methods inherently struggle to capture scene semantics and compositions, limiting their interpretability and interactivity. To address this limitation, we propose ObjLoc, a camera relocalization system designed to provide an intuition of scene object compositions and accurate pose estimation, which can be seamlessly reused in high-level tasks. Specifically, leveraging recent foundation models, we first introduce a multi-modal strategy to integrate open-vocabulary semantic knowledge for effective 2D-3D object matching. Additionally, we design an object-oriented reference frame and a corresponding retrieval strategy for pose priors, enabling extension to scalable scenes. To ensure robust and accurate pose optimization, we also propose a novel dual-path 2D Iterative Closest Pixel loss guided by object geometry. Experimental results demonstrate that ObjLoc achieves superior relocalization performance across various datasets. Our source code will be released upon acceptance.

## 1 INTRODUCTION

Indoor visual relocalization has been a fundamental problem in 3D computer vision over recent decades, especially with trending applications such as virtual/augmented reality (VR/AR), robot-environment interaction, and navigation, which targets estimating the 6-DOF camera pose given a sensor observation in a known map. At present, facing increasingly challenging requirements for embodied agents, an indoor relocalization system is desired to evolve beyond just accuracy, towards scalability, compactness, and most importantly, semantic awareness, to improve its versatility and adaptability in various downstream tasks such as robot-object interaction.

Previous visual relocalization solutions Kendall et al. (2015); Szegedy et al. (2015); Brahmbhatt et al. (2018); Kendall & Cipolla (2017); Camposeco et al. (2017); Cheng et al. (2019) mainly rely on low-level vision features and are limited by the absence of scene semantic understanding and memory-accuracy balance. As a result, they inevitably overlooked the inherent characteristics of indoor scenes and struggled to support high-level applications. By contrast, humans can relocalize themselves by recognizing surrounding objects, implying object elements can serve as valuable cues for relocalization. An indoor scene is essentially a spatial composition of 3D objects, characterized by rich semantics, regular geometry, and a distinct layout, which all concentrate at the object level. In addition, an object-oriented map typically collects basic geometry attributes (3D bounding boxes, colorless point clouds, *etc*.) of objects, thereby remaining highly compact. Meanwhile, objects are also fundamental units that can be interpreted and interacted with by a robot. Therefore, performing camera relocalization in an object-oriented map is highly promising, especially for indoor scenarios rich in diverse objects.

However, as an novel task setting, there are only a few emerging attempts Wang et al. (2024); Matsuzaki et al. (2024b) to exploit objects in 6-DOF camera relocalization, and they still stay in the preliminary stage. Specifically, existing works primarily suffer from the following three drawbacks: **(1)** Existing landmark association techniques suffer from non-discriminative and information-poor object descriptors, which may lead to severe outliers in object matching. **(2)** Reliable pose prior is necessary for relocalization in scalable indoor scenes, yet it is absent in existing object-level works. **(3)** Previous works often optimize poses via aligning 2D-3D bounding box centers, which results

Figure 1: **ObjLoc**, an object-level camera relocalization system, can achieve robust and accurate relocalization performance on various indoor scenes, based on an open-vocabulary object map. As shown in the figure, in an extremely large multi-floor scene, the robot observes a tiny corner containing a small radio and long-tailed animal ornament, and our system successfully identifies their 3D correspondences among hundreds of landmarks in the map. Next, initializing a pose prior, we can optimize the camera pose with a novel loss design.

in significant ambiguity and errors in the case of few object correspondences. A dedicated pose optimization technique tailored for object-level camera relocalization is still lacking.

To address the above challenges, we propose a semantic-aware, memory-efficient, and scalable indoor camera relocalization framework based on open-vocabulary object-level mapping, *i.e.*, **ObjLoc**. ObjLoc constructs a novel and well-designed object-oriented map suite that consists of a global scene graph, open-vocabulary object descriptors, object geometry, and reference frames. At first, by leveraging open-vocabulary descriptors and the global scene graph, ObjLoc enables robust multi-modal object matching to overcome the landmark association issue. Specifically, we employ an advanced off-the-shelf foundation model, CLIP Radford et al. (2021), to embed both visual and textual concepts into object descriptors. These descriptors capture high-level semantic knowledge such as affordance, material, *etc.*, enabling accurate recognition of class-agnostic objects. Meanwhile, the global scene graph can be utilized to incorporate the layout context as an informative modality to further improve landmark association. Secondly, to expand to scalable scenes, ObjLoc introduces object-oriented reference frames, a compact and efficient representation that records only observed object IDs and 2D bounding box coordinates instead of redundant appearance color. Based on this representation, a new DIoU-based (Distance-IoU) retrieval strategy is derived to measure frame similarity between query and database images, providing reliable pose priors. Finally, we propose an object-level tracker with a novel dual-path 2D ICP (Iterative Closest Pixel) loss, which estimates accurate camera poses by aligning observed and projected pixel areas of objects. This fine-grained loss can provide strong pose guidance even under sparse object correspondences. Benefiting from this innovation, ObjLoc achieves exceptional accuracy gain beyond existing works.

We evaluate our system on benchmark indoor datasets, including ScanNet Dai et al. (2017) and ScanNet++ Yeshwanth et al. (2023). Furthermore, we synthesized multiple large-scale scenes based on the Habitat Savva et al. (2019); Puig et al. (2023); Szot et al. (2021) simulator to cover a wide range of object categories. To the best of our knowledge, ObjLoc is the first object-level method that can work in such large-scale scenes. (See in Fig. 1). Extensive experiment results demonstrate that our ObjLoc outperforms existing approaches, yielding superior performance in visual relocalization. Overall, our contributions can be summarized as follows:

- We present a comprehensive framework that maintains object semantics, relocalization accuracy, and map compactness, offering a fresh perspective for indoor camera relocalization.

- We develop a novel and well-designed object-level map suite that enables robust multi-modal landmark association to ensure sufficient inliers, while also supporting coarse pose prior search in scalable scenes along with a DIOU-based retrieval strategy.

- We propose a novel dual-path 2D ICP loss tailored for pose optimization with object-level correspondences, which can significantly improve relocalization accuracy and robustness.

- Experiments on various datasets demonstrate that our object-level system consistently achieves state-of-the-art performance in indoor camera relocalization.

## 2 RELATED WORK

**Open-Vocabulary Semantic Mapping.** Traditional semantic mapping commonly trains a neural classifier on fixed object categories. While effective on known scenes and objects, they fail to generalize to long-tail categories and complex scenarios. Recent progress in 2D vision-language foundation models, such as Radford et al. (2021); Jia et al. (2021); Girdhar et al. (2023), has advanced the shift in semantic mapping from closed-set approaches to open-vocabulary ones. This enables robust zero-shot recognition and alleviates the need for labor-intensive annotations. OpenScene Peng et al. (2023) introduces the open-vocabulary scene mapping and understanding task. It directly projects 2D CLIP features onto dense 3D point clouds. However, point-wise feature alignment is prone to noisy and incomplete segmentation results. ConceptFusion Jatavallabhula et al. (2023) also builds a point-level map but enriches it with additional modalities. Takmaz et al. (2023); Nguyen et al. (2024) adopt a 3D model to first generate class-agnostic instance proposals, then extract instance-wise open-vocabulary features, which improves the accuracy of object recognition. Nevertheless, pre-trained 3D models struggle to ensure reliable segmentation performance. Following Yang et al. (2023); Yan et al. (2024); Yin et al. (2024); Lu et al. (2023) utilize powerful 2D segmentation models to produce 2D class-agnostic masks, and merge them into instances. Lifting 2D segmentations into 3D space can effectively enhance instance quality. The above works provide references for how to build an object-oriented map in an object-level relocalization system, but they lack relocalization task-oriented designs when handling challenging indoor situations, such as similar or repeated objects.

**Object-Level Relocalization.** Recently, object-level SLAM has gained widespread attention. By matching the mapping frames with 3D instances and minimizing the projection error, object-level SLAM Salas-Moreno et al. (2013); Yang & Scherer (2019b); Zins et al. (2022b); Yang & Scherer (2019a); Wu et al. (2020); Wang et al. (2021) shows satisfactory pose results. Although object-based SLAM methods have been researched, object-based relocalization remains relatively underexplored. Zins et al. (2022a) first proposes an object-level relocalization pipeline by matching the object category in the query and the pre-built map. However, object-level matching is prone to incorrect associations, which may lead to degradation in relocalization accuracy. GOReloc Wang et al. (2024) considers the semantic uncertainty and consistency in a graph to facilitate object matching. However, such graph-based object descriptors only record close-vocabulary neighbor categories and numbers and thus remain confused and non-discriminative in the association. Clip-Loc Matsuzaki et al. (2024a) first tried to introduce open-vocabulary features as object descriptors, and Clip-Clique Matsuzaki et al. (2024b) further proposed to combine maximal clique finding with CLIP to improve matching performance. Unfortunately, they lacked a complete map suite and a systematic pipeline, which prevented them from fully exploring the potential of this research line. This is also the reason for their limited performance and scalability.

## 3 METHOD

***Problem Statement.*** The overview of our proposed object-level camera relocalization framework is shown in Fig 2. Given a collection of posed RGBD images from a scene, and an unseen query RGB image from the same scene, the topic of the object-level camera relocalization task is to estimate the 6-DoF camera pose of this query image solely based on key attributes of scene objects, such as high-level semantics, neighbor relationships, and geometric shapes.

***Step-by-step Overview.*** **(1)** Object-oriented Mapping (Sec 3.1): Given a set of posed RGBD images from a scene, the task is to process these RGBD observations and output an object-centric 3D map suite, including a 3D instance segmented point cloud, per-object feature descriptor, object-oriented reference frames, and a global scene graph. **(2)** Landmark Association (Sec 3.2): Given a pre-built map from the last step and an unseen RGB query image, we analyze this query image and find correspondences between observed objects in the query image and those objects (landmarks) in the 3D map. **(3)** Relocalization (Sec 3.3): Given the 3D map, 2D query image, and object matching pairs between them, we employ a coarse-to-fine strategy to accurately and robustly estimate the 6-DoF camera pose of this query image.

### 3.1 OBJECT-ORIENTED MAPPING

Object-oriented mapping is the first and pivotal step in our framework, where a well-structured map suite and high-quality reconstruction serve as the core foundation. In this section, we introduce a sequential object-level mapping pipeline and the principles behind each module.

Figure 2: **System Overview.** As shown in this figure, our system includes three main steps: (1) We lift a RGBD-S (RGBD-Segmentation) sequence into 3D space to obtain object landmarks $O_l$ and reference frames $\mathcal{K}$. Well visible patches $\mathcal{S}$ of a landmark at different views are fed into CLIP for its descriptor $f^{3d}$. Finally, based on 3D bounding box collisions and nearest neighbors, we derive a global scene into a graph $\mathcal{G}$. (2) In a query image, detected object regions and their descriptions from LLM are both fed into CLIP for query features $f_{vision}^{2d}, f_{text}^{2d}$. We distinguish similar objects by the scene graph analysis, *i.e.*, layout context. (3) DIOU-based retrieval can find a reference frame most similar to the query image for initialization. Then, leveraging the geometric shapes of objects, the camera pose is optimized under the guidance of aligning the projected and target pixel areas.

***Instance Segmentation.*** In the object-level relocalization task, instance segmentation plays an important role in identifying individual objects. Based on depth observations, we can reconstruct the scene mesh by TSDF-Fusion Zeng et al. (2017) and convert vertices into the scene point cloud $\mathcal{P}$. Then, CropFormer Qi et al. (2022) is utilized to predict 2D mask proposals on input RGB images as graph nodes $\mathcal{V}_m$. Edge affinity $\mathcal{E}_m$ is calculated with a simplified view consensus rate proposed in Yan et al. (2024). Through progressive graph clustering in $\mathcal{G}(\mathcal{V}_m|\mathcal{E}_m)$, we can merge mask nodes into clusters, each representing an instance. Instance segmentation module lifts 2D segmentations into 3D space and generates landmarks $\{O_l^i = (P_i, B_i^{3d}, C_i)|i = 1, 2, 3..N\}$ including point clouds $P_i$, 3D bounding boxes $B_i^{3d}$, and centers $C_i$. Some qualitative results are presented in the F.

***Distribution of Reference Frames.*** In order to adapt to scalable scenes, we innovatively create object-oriented reference frames and distribute them as initial pose anchors. Unlike the RGB reference frame, we replace the redundant color with landmark IDs $i$ and 2D bounding box coordinates $B_i^{2d}$ in our design. We define our reference frames $\mathcal{K}$ as Eq. equation 1 and visualize them in Fig. 2.

$$\mathcal{K} = \{(i, B_i^{2d})|i = 1, 2, .., N_\mathcal{K}\} \, , \tag{1}$$

where $N_\mathcal{K}$ represents the number of objects observed in $\mathcal{K}$. We select reference frames based on detected object difference, *e.g.*, a fresh object is observed or a historical object becomes much more visible. Such object-oriented reference frames effectively alleviate storage pressure caused by color images, especially in large-scale indoor scenarios.

***Multi-view Object Descriptor.*** Very recently, the advanced CLIP model can work as an effective object descriptor encoder. We project point clouds $P_i$ as point prompts in SAM Kirillov et al. (2023) to find patches $\mathcal{S}$ of the same landmark in different views. Top-$k$ segmentation patches with maximal visibility are input into a CLIP visual encoder and an average pooling layer to obtain a multi-view CLIP feature $f^{3d}$:

$$f_i^{3d} = \frac{1}{k} \sum_{n=1}^{k} \text{CLIP}(\mathcal{S}_n) \, . \tag{2}$$

These discriminative features $f^{3d}$, rich in open-vocabulary semantics, can serve as descriptors of class-agnostic objects. Notably, we slightly scale up each patch to involve some surroundings.

***Invalid Object Recognition*** We have noticed that some indoor objects do not carry valuable information and still occupy a large portion of the map, such as the wall and floor. We consider these objects to be invalid or even negative, and it is necessary to ignore them in subsequent steps. To avoid additional computational complexity, we recognize these objects based on their frequency in existing reference frames. If an object appears in the majority of reference frames, it cannot assist our system in narrowing down the pose retrieval region. Qualitative results are presented in D.

***Scene Graph Extraction.*** We believe that spatial arrangement is an essential cue for indoor object-level relocalization. As an interpretable representation, a scene graph can clearly describe object relationships and the contextual layout to disambiguate the landmark association. In our global scene graph $\mathcal{G}(\mathcal{V}|\mathcal{E})$, edges $\mathcal{E}$ are added between each object node $\mathcal{V}$ with its nearest neighbors or those objects exhibiting 3D bounding box collisions.

## 3.2 Landmark Association

Landmark association refers to a process where our system searches for observed 2D objects in the query image within a pre-built object-level 3D map. This has historically been challenging, especially in large-scale scenes. In this section, we discussed how to establish 2D-3D object correspondences in terms of vision, language, and layout context.

***Open-vocabulary Matching.*** In a query image, to fully discover objects in it, we adopt the powerful Florence2 Xiao et al. (2024) model to perform object region proposal. These found object regions (2D bounding boxes) are fed into CLIP to obtain visual features $f^{2d}_{vision}$. We can quantify the uniqueness $\gamma$ of a region with its cosine similarity variance with all landmarks, as shown in Eq. equation 3.

$$\gamma = Var(cos(f^{2d}_{vision}, f^{3d}_1), ..., cos(f^{2d}_{vision}, f^{3d}_N)) , \qquad (3)$$

where $Var(\cdot)$ is the variance and $cos(\cdot)$ is the cosine similarity. Only 2D Object regions with sufficient uniqueness will be retained for subsequent recognition and matching.

Next, we hope to automatically annotate text descriptions on these salient 2D objects $O_q$. Large Language Models (LLMs) have exhibited remarkable common-sense reasoning ability. This is the reason why we use GPT-4o Hurst et al. (2024) as an agent for object analysis. Nevertheless, we still need to craft a reasonable LLM prompt to raise its task orientation and response quality. It is well known that the placement environment of an object is closely related to its functionality. However, too many surroundings may lead to visual interference, which affects the agent's inference. Consequently, as shown in Fig. 2, we simultaneously pass two images into the agent: the query image and a segmented 2D object. Our agent should pay attention to both surroundings and the object itself. We similarly fed agent responses to the CLIP for language descriptors $f^{2d}_{text}$. We can now correlate query features $f^{2d}_{vision}, f^{2d}_{text}$ with $f^{3d}$ for Top-3 landmark candidates $O_{vis}, O_{text}$ respectively:

$$cos(f^{2d}_{vision}, f^{3d}_{i=1,..,N}) \Rightarrow O_{vis}=\{O_l^{v1}, O_l^{v2}, O_l^{v3}\} , \qquad (4a)$$

$$cos(f^{2d}_{text}, f^{3d}_{i=1,..,N}) \Rightarrow O_{text}=\{O_l^{t1}, O_l^{t2}, O_l^{t3}\} . \qquad (4b)$$

If visual and language cues indicate the same 3D object ($O_l^{v1} = O_l^{t1}$), we regard it as confident enough to be inserted into the final landmark association results $L$. Otherwise, we give a set of 3D object candidates $U=\{(O_{vis} \cap O_{text}) \cup O_l^{v1} \cup O_l^{t1}\}$ to be further checked in scene graph analysis.

***Sub-graph Matching.*** As in Fig. 2, frequent repeated or similar objects in an indoor scene are the root cause of the uncertainty set $U$. We resort to scene graph analysis to address this problem by leveraging neighborhood relationships. For a 2D object in the query image, its subgraph $\mathcal{G}_q$ will be constructed by connecting the nearest or other intersecting 2D object regions. For a candidate in $U$, it is regarded as an origin in the global scene graph $\mathcal{G}(\mathcal{V}|\mathcal{E})$ to extract a Breadth-First 3D subgraph $\mathcal{G}_l$ with a path length $\eta$. We can determine the best candidate from $U$ by identifying a 3D subgraph $\mathcal{G}_l$ that is most similar to the $\mathcal{G}_q$.

Another problem is how to measure subgraph similarity, which is formulated as a linear sum assignment problem (LSAP). The goal is to solve an optimal neighbor nodes assignment from $\mathcal{G}_q$ to

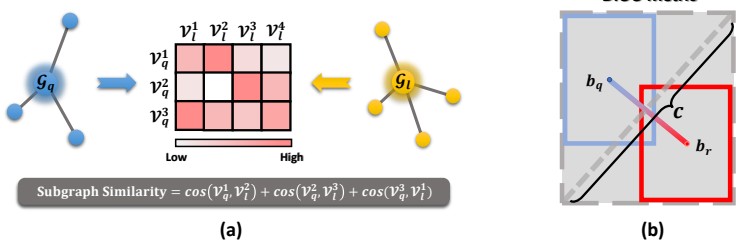

Figure 3: **Subgraph Similarity and DIOU Metric.** (a) We compute the cosine similarity of all possible neighbor pairs between two subgraphs, from which we solve an assignment to maximize the total matching score as the subgraph similarity. (b) This figure visualizes relevant variables $b_q, b_r, c$ and the principle of DIOU metric.

$\mathcal{G}_l$ so that the total matching score is maximized, as shown in Fig. 3 (a). After filtering via subgraph matching, we can obtain the final landmark association results:

$$L = \{(O_q^{i_1}, O_l^{i_2})|i_1 \in (1, .., N_q), i_2 \in (1, ...N)\} . \tag{5}$$

### 3.3 RELOCALIZATION

As the final pose estimation step, the object-level tracker directly influences the relocalization success rate and accuracy. Our object-level tracker improves relocalization performance relying on a coarse-to-fine strategy and a novel loss. Two relocalization stages are detailed below.

***Coarse Pose Prior.*** For voting a reference frame most similar to the query image, we initially choose the ones that contain the largest number of matched 3D landmarks. This often results in a co-visible subset of reference frames, for which a DIOU metric is further calculated as follows:

$$\text{DIOU} = 1 - IOU + \frac{||\mathbf{b_q} - \mathbf{b_r}||^2}{c^2} , \tag{6}$$

where $\mathbf{b_q}$ and $\mathbf{b_r}$ represent 2D bounding box centers of the same object in the query and reference frames respectively, and $c$ is the diagonal distance of the smallest enclosing rectangle covering two boxes, as shown in Fig. 3 (b). This metric is compatible with our reference frame design and it can avoid failure cases from non-overlapping boxes (IOU=0). The lower the DIOU metric, the better. So far, the reference frame $\mathcal{K}$ with the best DIOU score offers a coarse pose prior.

***Refined Pose Optimization.*** Previous object-level works always try to optimize poses by aligning the center points of 2D and 3D bounding boxes. Although this approach can provide a rough optimization trend, it is inherently unsuitable for an object-level system. In principle, it is prone to severe errors or ambiguity due to sparse object center point correspondences. Inspired by the traditional 3D ICP (Iterative Closest Point) algorithm, we evolved it into a 2D ICP variant on the image plane to optimize camera poses $\{q, T\}$, *i.e.*, quaternion rotation and translation. Specifically, as shown in Fig. 2, given the point cloud $P_i$ of a landmark and its target mask area $m_i$ in the query. We can project $P_i$ into a pixel set $p_i$ at the current pose and compute the bidirectional average distance between the closest pixel pairs in $p_i$ and $m_i$, as shown in Eqs. 7a and 7b. To enable more robust and accurate pose estimation, we impose a Huber kernel $\mathcal{H}$ with a hyperparameter $\delta$ on the 2D ICP loss to suppress extreme outlier pixels.

$$\mathcal{L}_{forward}^i = \frac{1}{N_{p_i}} \sum_{n \in p_i} \mathcal{H}(||p_i^n - \psi(p_i^n, m_i)||^2, \delta) , \tag{7a}$$

$$\mathcal{L}_{backward}^i = \frac{1}{N_{m_i}} \sum_{n \in m_i} \mathcal{H}(||m_i^n - \psi(m_i^n, p_i)||^2, \delta) , \tag{7b}$$

$$\mathcal{L}_{icp} = \frac{1}{N_L} \sum_{i \in L} (\mathcal{L}_{forward}^i + \mathcal{L}_{backward}^i) , \tag{8}$$

where $N_{p_i}$, $N_{m_i}$ are the number of pixels in $p_i$ and $m_i$, $N_L$ is the number of matching objects, and $\psi(\cdot)$ outputs the closest peer of a pixel. Our dual-path 2D ICP loss in Eq. 8 can make full use of object information to align not only centers of objects but also their entire 2D shapes. The importance of bidirectional design lies in eliminating scale ambiguity arising from $\mathcal{L}_{forward}$ or $\mathcal{L}_{backward}$ alone. Benefiting from this loss, we can achieve stable and accurate pose optimization.

Table 1: **Relocalization Results on the ScanNet Dataset.** Our system achieves the state-of-the-art relocalization performance in terms of recall and accuracy in the ScanNet dataset.

| Method | Metric | 0568 | 0101 | 0673 | 0108 | 0166 | 0378 | 0092 | 0603 | Method | Metric | 0568 | 0101 | 0673 | 0108 | 0166 | 0378 | 0092 | 0603 |
|---|---|---|---|---|---|---|---|---|---|---|---|---|---|---|---|---|---|---|---|
| | | | | @50cm | | | | | | | | | | @25cm | | | | | |
| CoordiNet | Recall[%]↑ | 36 | 17 | 32 | 18 | 61 | 44 | 38 | 33 | CoordiNet | Recall[%]↑ | 6 | 3 | 8 | 4 | 21 | 17 | 6 | 9 |
| | MTE[m]↓ | 0.34 | 0.35 | 0.32 | 0.36 | 0.31 | 0.32 | 0.35 | 0.32 | | MTE[m]↓ | 0.21 | 0.21 | 0.18 | 0.23 | 0.17 | 0.22 | 0.18 | 0.17 |
| | MRE[°]↓ | 13.6 | 17.8 | 10.5 | 21.6 | 11.3 | 14.8 | 14.5 | 11.3 | | MRE[°]↓ | 19.2 | 8.2 | 9.0 | 31.1 | 11.4 | 14.1 | 8.2 | 12.5 |
| MS-Transformer | Recall[%]↑ | 76 | 37 | 14 | 82 | 52 | 41 | 45 | 40 | MS-Transformer | Recall[%]↑ | 32 | 14 | 4 | 57 | 33 | 21 | 19 | 9 |
| | MTE[m]↓ | 0.28 | 0.31 | 0.32 | 0.21 | 0.23 | 0.26 | 0.25 | 0.32 | | MTE[m]↓ | 0.16 | 0.18 | 0.23 | 0.15 | 0.17 | 0.16 | 0.14 | 0.16 |
| | MRE[°]↓ | 23.2 | 43.4 | 46.8 | 41.7 | 27.4 | 33.4 | 15.3 | 25.5 | | MRE[°]↓ | 18.6 | 35.8 | 9.4 | 31.5 | 21.2 | 34.2 | 12.7 | 22.0 |
| GoReloc | Recall[%]↑ | 8 | 17 | 22 | 25 | 16 | 12 | 9 | 14 | GoReloc | Recall[%]↑ | 5 | 17 | 10 | 12 | 7 | 3 | 6 | 5 |
| | MTE[m]↓ | 0.23 | 0.21 | 0.29 | 0.26 | 0.23 | 0.30 | 0.23 | 0.29 | | MTE[m]↓ | 0.14 | 0.21 | 0.14 | 0.12 | 0.13 | 0.16 | 0.14 | 0.14 |
| | MRE[°]↓ | 4.6 | 4.9 | 10.2 | 4.5 | 7.0 | 9.5 | 8.6 | 26.8 | | MRE[°]↓ | 2.1 | 4.9 | 4.5 | 3.7 | 4.9 | 5.2 | 7.4 | 6.1 |
| **Ours** | Recall[%]↑ | 79 | 68 | 64 | 89 | 72 | 83 | 69 | 65 | **Ours** | Recall[%]↑ | 58 | 45 | 51 | 73 | 52 | 74 | 50 | 52 |
| | MTE[m]↓ | 0.18 | 0.2 | 0.17 | 0.16 | 0.18 | 0.12 | 0.19 | 0.15 | | MTE[m]↓ | 0.13 | 0.14 | 0.12 | 0.11 | 0.12 | 0.09 | 0.12 | 0.10 |
| | MRE[°]↓ | 4.0 | 4.7 | 4.6 | 3.9 | 6.0 | 3.6 | 6.0 | 5.7 | | MRE[°]↓ | 2.9 | 3.6 | 3.4 | 2.8 | 4.0 | 2.7 | 4.0 | 3.8 |

Table 2: **Relocalization Results on the ScanNet++ Dataset.** From this table, it is clear that our system achieves a better relocalization success rate and lower translation/rotation errors.

| Method | Metric | 0a7cc | 0a184 | 0d2ee | 7e094 | 7f4d1 | 25f3b | 8890d | a08d9 | Method | Metric | 0a7cc | 0a184 | 0d2ee | 7e094 | 7f4d1 | 25f3b | 8890d | a08d9 |
|---|---|---|---|---|---|---|---|---|---|---|---|---|---|---|---|---|---|---|---|
| | | | | @50cm | | | | | | | | | | @25cm | | | | | |
| CoordiNet | Recall[%]↑ | 35 | 39 | 32 | 62 | 36 | 54 | 64 | 35 | CoordiNet | Recall[%]↑ | 10 | 7 | 5 | 21 | 9 | 15 | 13 | 11 |
| | MTE[m]↓ | 0.31 | 0.35 | 0.37 | 0.33 | 0.33 | 0.32 | 0.32 | 0.34 | | MTE[m]↓ | 0.18 | 0.16 | 0.20 | 0.21 | 0.16 | 0.19 | 0.15 | 0.19 |
| | MRE[°]↓ | 13.3 | 13.4 | 26.1 | 6.2 | 11.3 | 10.1 | 6.4 | 11.8 | | MRE[°]↓ | 14.0 | 12.2 | 22.5 | 5.9 | 13.1 | 8.2 | 6.5 | 13.7 |
| MS-Transformer | Recall[%]↑ | 68 | 72 | 32 | 60 | 57 | 60 | 66 | 51 | MS-Transformer | Recall[%]↑ | 41 | 32 | 11 | 35 | 18 | 40 | 27 | 11 |
| | MTE[m]↓ | 0.25 | 0.26 | 0.31 | 0.22 | 0.29 | 0.22 | 0.28 | 0.32 | | MTE[m]↓ | 0.17 | 0.16 | 0.17 | 0.13 | 0.16 | 0.16 | 0.17 | 0.18 |
| | MRE[°]↓ | 14.2 | 26.8 | 39.8 | 38.2 | 25.4 | 22.8 | 26.3 | 18.7 | | MRE[°]↓ | 12.2 | 25.3 | 26.0 | 13.1 | 23.4 | 20.2 | 17.3 | 12.1 |
| **Ours** | Recall[%]↑ | 70 | 74 | 62 | 92 | 75 | 77 | 81 | 70 | **Ours** | Recall[%]↑ | 60 | 59 | 54 | 80 | 68 | 65 | 70 | 54 |
| | MTE[m]↓ | 0.11 | 0.12 | 0.13 | 0.11 | 0.09 | 0.13 | 0.09 | 0.16 | | MTE[m]↓ | 0.07 | 0.07 | 0.10 | 0.08 | 0.06 | 0.09 | 0.06 | 0.09 |
| | MRE[°]↓ | 4.0 | 4.1 | 4.2 | 3.7 | 3.1 | 7.1 | 3.7 | 8.0 | | MRE[°]↓ | 2.7 | 2.5 | 3.7 | 2.4 | 2.6 | 5.1 | 2.0 | 4.0 |

# 4 EXPERIMENTS

In this section, we describe our experimental setup and validate that our system can achieve significant improvements in relocalization performance. We evaluate our system in single-room (Sec. 4.1), multi-room (Sec. 4.2), and even multi-floor indoor environments. We analyze its map size (Sec. 4.3) and confirmed the effectiveness of key module designs (Sec. 4.4). We color each cell as **best** , second best , and third best .

**Datasets.** In the single-room case, we utilize two real-world benchmarks: ScanNet Dai et al. (2017) and ScanNet++ Yeshwanth et al. (2023), which contain 8 challenging scenes respectively. To acquire query images, in these two datasets, we split a continuous image sequence into two parts for mapping and relocalization. In the multi-room case, for free query viewpoints and rich objects, we use the Habitat simulator Savva et al. (2019); Szot et al. (2021); Puig et al. (2023) to generate 6 multi-room and 2 multi-floor scenes based on available assets from the HSSD Khanna* et al. (2023).

**Implementation Details.** We run our system on a desktop equipped with an Intel i9-14900K and an NVIDIA RTX 4090 GPU. We set the learning rate of $\{q, T\}$ to $\{0.025, 0.025\}$ in the refined pose optimization. During extracting 3D landmark descriptors, we select $k=5$ segmentation patches. In 2D ICP loss, we use the hyperparameter $\delta = 10$ in the Huber kernel. The path length $\eta$ is set to $\eta = 1$ in the node search to extract 3D subgraphs from the global scene graph. Our system selects the official ViT-L/14@336px CLIP model to embed 768-dimension open-vocabulary features. In ScanNet Dai et al. (2017), ScanNet++ Yeshwanth et al. (2023), and Synthetic datasets, we sample mapping frames at intervals of $\{20, 40, 10\}$. We sample query frames starting from the 10th frame with intervals of $\{20, 80\}$ in ScanNet and ScanNet++. For further implementation details, please refer to A.

**Metrics.** A relocalization system is most concerned about its success rate and pose accuracy. We quantitatively evaluate these two aspects using different metrics. With respect to success rate, we count the percentage of correctly relocalized query images within given translation thresholds: 50cm and 25cm, *i.e.*, $Recall[\%]$ at 50cm and 25cm. As for accuracy, we calculated the mean translation error ($MTE[cm]$) and mean rotation error ($MRE[°]$) for those query images within $Recall@50cm$ and $Recall@25cm$, respectively.

**Baselines.** As a novel task setting, ObjLoc should be compared with some object-level 6-DOF relocalization methods that similarly possess semantic awareness. However, very limited open-source work is available in this emerging research field. Therefore, we primarily compare our method with an open-source and SOTA object-level baseline, GoReloc Wang et al. (2024), which shares the most relevant problem formulation with ours. Additionally, we also include some low-level vision methods Moreau et al. (2022); Shavit et al. (2021) with comparable map sizes for completeness and fairness . We reproduce and fine-tune all baselines on our own datasets and experimental settings.

Table 3: **Relocalization Results on the Synthetic Dataset.** Our system outperforms other baselines across various large-scale scenes. '-' denotes failure results in GoReloc Wang et al. (2024). '*' denotes an additional experiment group.

| Method | Metric | Scene1 | Scene2 | Scene3 | Scene4 | Scene5 | Scene6 | Scene7 | Scene8 |
|---|---|---|---|---|---|---|---|---|---|
| | | @50cm | | | | | | | |
| CoordiNet | Recall[%]↑ | 7 | 13 | 7 | 8 | 10 | 15 | 3 | 9 |
| | MTE[m]↓ | 0.31 | 0.32 | 0.43 | 0.27 | 0.33 | 0.38 | 0.27 | 0.37 |
| | MRE[°]↓ | 14.1 | 13.1 | 33.7 | 17.4 | 18.9 | 15.3 | 11.9 | 11.7 |
| MS-Transformer | Recall[%]↑ | 13 | 30 | 34 | 53 | 28 | 35 | 16 | 27 |
| | MTE[m]↓ | 0.33 | 0.33 | 0.34 | 0.31 | 0.28 | 0.35 | 0.33 | 0.31 |
| | MRE[°]↓ | 4.0 | 6.2 | 4.5 | 7.4 | 5.6 | 7.8 | 4.1 | 7.2 |
| GoReloc | Recall[%]↑ | 7 | 8 | 0 | 0 | 21 | 29 | - | - |
| | MTE[m]↓ | 0.41 | 0.38 | - | - | 0.21 | 0.35 | - | - |
| | MRE[°]↓ | 10.6 | 14.6 | - | - | 5.8 | 12.7 | - | - |
| **Ours** | Recall[%]↑ | 86 | 88 | 89 | 90 | 91 | 86 | 79 | 83 |
| | MTE[m]↓ | 0.12 | 0.07 | 0.11 | 0.14 | 0.09 | 0.09 | 0.1 | 0.09 |
| | MRE[°]↓ | 3.8 | 2.4 | 4.7 | 5.4 | 4.1 | 4.2 | 3.4 | 3.6 |

| Method | Metric | Scene1 | Scene2 | Scene3 | Scene4 | Scene5 | Scene6 | Scene7 | Scene8 |
|---|---|---|---|---|---|---|---|---|---|
| | | @25cm | | | | | | | |
| MS-Transformer | Recall[%]↑ | 4 | 8 | 7 | 20 | 13 | 4 | 5 | 9 |
| | MTE[m]↓ | 0.16 | 0.21 | 0.17 | 0.18 | 0.14 | 0.16 | 0.16 | 0.17 |
| | MRE[°]↓ | 4.9 | 6.8 | 3.6 | 3.5 | 4.7 | 14.1 | 3.6 | 3.8 |
| **Ours** | Recall[%]↑ | 73 | 83 | 75 | 69 | 86 | 74 | 71 | 75 |
| | MTE[m]↓ | 0.07 | 0.05 | 0.06 | 0.07 | 0.06 | 0.05 | 0.06 | 0.06 |
| | MRE[°]↓ | 1.8 | 1.9 | 2.3 | 2.6 | 2.7 | 2.0 | 2.2 | 2.5 |
| | | *@100cm | | | | | | | |
| CoordiNet* | Recall[%]↑ | 19 | 46 | 29 | 32 | 33 | 40 | 24 | 45 |
| | MTE[m]↓ | 0.58 | 0.62 | 0.70 | 0.61 | 0.60 | 0.63 | 0.68 | 0.68 |
| | MRE[°]↓ | 29.7 | 15.9 | 24.8 | 20.6 | 18.3 | 17.9 | 15.6 | 13.9 |
| GoReloc* | Recall[%]↑ | 11 | 17 | 10 | 25 | 21 | 35 | - | - |
| | MTE[m]↓ | 0.58 | 0.54 | 0.59 | 0.56 | 0.21 | 0.48 | - | - |
| | MRE[°]↓ | 11.5 | 18.0 | 12.5 | 33.2 | 5.8 | 18.2 | - | - |

(a) ScanNet      (b) ScanNet++      (c) Synthetic

Figure 4: **Relocalization Results on Various Datasets.** In this figure, we qualitatively show some relocalization poses of our system and their ground truth on various datasets. It is obvious that our system can achieve accurate relocalization in scalable scenes.

### 4.1 SINGLE-ROOM RELOCALIZATION

**Evaluation on ScanNet Dai et al. (2017).** We report relocalization results of our system across 8 ScanNet scenes in Tab. 1. Despite suffering from incomplete and noisy RGB-D observations, excellent landmark association and innovative pose refinement promote a robust convergence of our system to a decent solution. It is clear that our system surpasses all other baselines with a notable margin in both success rate and accuracy at *Recall@50cm* and *Recall@25cm*. Against GoReloc Wang et al. (2024), our method substantially increases the success rate by around **5∼10** times. This breakthrough means that our system sets an upgraded benchmark for the object-level camera relocalization task.

**Evaluation on ScanNet++ Yeshwanth et al. (2023).** As shown in Tab. 2, we also evaluate the relocalization performance of our system on ScanNet++. In GoReloc Wang et al. (2024), its mapping relies heavily on the combination of YOLOv8 Redmon et al. (2016) and ORB-SLAM2 Mur-Artal & Tardós (2017) features, and weak-texture conditions in ScanNet++ scenes seriously hinder its normal operation. As a result, we only compare low-level vision approaches to ours in this dataset. Similarly, our system can still achieve better relocalization performance, and the availability of high-quality sensor data in ScanNet++ facilitates higher pose accuracy (*MTE* and *MRE*) of our system.

The poor performance of GoReloc in sing-room scenes primarily stems from its closed-vocabulary semantics, ambiguous pose optimization method, and low-discriminability descriptors that only encode neighbor categories. To sum up, extensive experiments on these two datasets illustrate the strong ability and open-vocabulary advantages of our system to handle complex real-world scenes, enhancing the practicality of object-level camera relocalization.

### 4.2 MULTI-ROOM RELOCALIZATION

With available assets and stages in the HSSD Khanna* et al. (2023), we assembled a variety of more challenging indoor scenes, consisting of multi-room (Scene1∼6) and multi-floor (Scene7∼8) cases to fully explore the potential of our system in terms of generalization and scalability. In Tab. 3, we quantitatively compare our system with baselines on the synthetic dataset. It can be seen that our system exhibits superior robustness in this large-scale setting, excelling other baselines in all metrics. These baselines struggle with large-scale scenes, experiencing a significant drop in their performance. Especially for GoReloc Wang et al. (2024) and CoordiNet Moreau et al. (2022), in such challenging scenarios, their *Recall@50cm* is very poor, and they fail to localize the camera pose within 25cm. Thus, we additionally add a more lenient condition, *Recall@100cm*, to examine them.

Multi-room experiments reveal that our framework can effortlessly adapt to large-scale indoor scenes, a capability we attribute to our object-oriented reference frame and DIOU retrieval strategy. In contrast, the lack of pose priors severely limits the scalability of GoReloc.

Table 5: **Ablation Study.** To validate the effectiveness of our main module designs, we report ablation results on scenes from different datasets [**0568** | **0a7cc** | **Scene1**]. '**-**' denotes failure cases.

| Setting | @50cm | | | | | | | | | @25cm | | | | | | | | |
|---|---|---|---|---|---|---|---|---|---|---|---|---|---|---|---|---|---|---|
| | Recall[%]↑ | | | MTE[m]↓ | | | MRE[°]↓ | | | Recall[%]↑ | | | MTE[m]↓ | | | MRE[°]↓ | | |
| #1 w/o Refine Stage | 62 | 45 | 6 | 0.25 | 0.15 | 0.43 | 8.5 | 11.0 | 29.7 | 46 | 41 | 0 | 0.18 | 0.09 | - | 6.1 | 11 | - |
| #2 w/o Coarse Stage | 27 | 0 | 29 | 0.29 | - | 0.17 | 8.3 | - | 5.0 | 11 | 0 | 20 | 0.15 | - | 0.09 | 3.3 | - | 2.9 |
| #3 w/o Scene Graph | 64 | 64 | 78 | 0.21 | 0.11 | 0.12 | 4.6 | 4.6 | 3.5 | 41 | 56 | 70 | 0.14 | 0.08 | 0.08 | 3.1 | 4.0 | 2.1 |
| #4 w/o Language Modality | 66 | 52 | 76 | 0.18 | 0.12 | 0.13 | 4.1 | 4.4 | 4.0 | 49 | 44 | 62 | 0.13 | 0.07 | 0.07 | 2.9 | 2.8 | 1.8 |
| **Ours Full** | **79** | **70** | **86** | **0.18** | **0.11** | **0.12** | **4.0** | **4.0** | **3.8** | **58** | **60** | **73** | **0.13** | **0.07** | **0.07** | **2.9** | **2.7** | **1.8** |

As shown in Fig. 4, we visualize relocalization results on some scenes of different datasets, which qualitatively demonstrate the effectiveness and generalization of our proposed framework. Notably, ObjLoc can recognize and match various objects in an open-vocabulary manner, which essentially sets ours apart from GoReloc and low-level methods. Hence, we conducted a targeted evaluation on this ObjLoc's particular strength on open-vocabulary object sets as in B. We also provide more visualizations, lighting variance evaluation, and further system analysis in our appendix material.

### 4.3 MAP SIZE ANALYSIS

We report the map size of different methods on the ScanNet '0568' scene in Tab. 4, where object-level methods (GoReloc Wang et al. (2024) and Ours) can construct a significantly

Table 4: **Map Sizes of Different Methods.**

| Metric | CoordiNet | MS-Transformer | GoReloc | **Ours** |
|---|---|---|---|---|
| Map Size[MB]↓ | 71.4 | 63.1 | 17.2 | **3.5** |

more lightweight map compared to low-level methods (CoordiNet Moreau et al. (2022) and MS-Transformer Shavit et al. (2021)). Furthermore, compared to GoReloc Wang et al. (2024), removing object color and category likelihood saves an additional **80%** of memory consumption in our system.

### 4.4 ABLATION STUDY

To verify the rationality of our main module designs, we conduct ablation studies on scenes from different datasets (0568, 0a7cc, and Scene1). We investigate the effectiveness of the coarse-to-fine pose optimization, scene graph analysis, and language modality. Please refer to D for more ablations on DIOU-based retrieval, invalid object filtering, and relocalization losses.

**Effectiveness of Coarse-to-fine Pose Optimization.** The coarse-to-fine pose optimization is a reasonable and necessary strategy in camera relocalization, especially for large-scale scenes. To demonstrate the necessity and effectiveness of coarse pose priors and pose refinement, we separately conducted ablation studies on them in Tab. 5 **#1#2**. When ablating coarse priors, we use the average of all ground-truth poses as a fixed initial value. Results show that removing either component severely degrades performance: without coarse priors, poor initialization disrupts refinement, causing the system to fail; without refinement, accuracy drops significantly. These results highlight the contributions and complementary roles of coarse and refinement stages, which jointly ensure accuracy.

**Effectiveness of Scene Graph.** Scene graph analysis aims to address uncertain candidates in $U$. Without scene graph analysis, the system is prone to being confused by similar or repeated objects, thereby weakening its object recognition capability. As shown in Tab. 5 **#3**, if the scene graph module is ablated, an obvious performance degradation will happen due to object mismatches.

**Effectiveness of Language Modality.** In Tab. 5 **#4**, we compare the system performance with and without language modality. In the presence of visual occlusion and noise, relying solely on visual cues to predict 3D object candidates is biased. However, text descriptions obtained from the common-sense reasoning of an LLM model can fix this weakness. Ablation results indicate that the language modality can effectively improve relocalization performance.

## 5 CONCLUSION

We have proposed ObjLoc, a comprehensive indoor camera relocalization system based on open-vocabulary object-level mapping, which is the first to handle scalable scenes at the object level. Multi-modal analysis for objects supports our system in better associating observed items in the query image with correct landmarks. Object-oriented reference frames and the DIOU-based retrieval address the scalability bottleneck, providing reliable pose priors. Besides, the dual-path ICP design effectively optimizes poses with sparse object correspondences, improving relocalization recall and accuracy. Through extensive experiments, it can be concluded that our system achieves pioneering progress and state-of-the-art performance in the object-level camera relocalization task.

**Limitations.** Our system cannot capture and locally update dynamic changes in a scene, which may cause some inconvenience. It is an interesting direction for future work.

## ETHICS STATEMENT

Our study focuses on visual camera relocalization, a core problem in the field of 3D vision. The experimental evaluation is based on public datasets and our self-synthesized datasets that were curated without the inclusion of sensitive content. We assert that this research has been carried out in accordance with the code of ethics.

## REPRODUCIBILITY STATEMENT

To ensure reproducibility and verification of our work, we include the implementation details and evaluation procedures in our paper, and will make our source code publicly available upon acceptance.

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

APPENDIX

In this appendix, we provide more details of our ObjLoc system, including 1) further implementation details (A); 2) open-vocabulary object matching evaluation (B); 3) lighting variance evaluation of our system under different lighting conditions (C); 4) ablation studies on the invalid object filtering, DIOU-based retrieval, and relocalization losses (D); 5) further system analysis (E); 6) more visualization results (F). We also provide a video attachment to visualize our framework and experimental results.

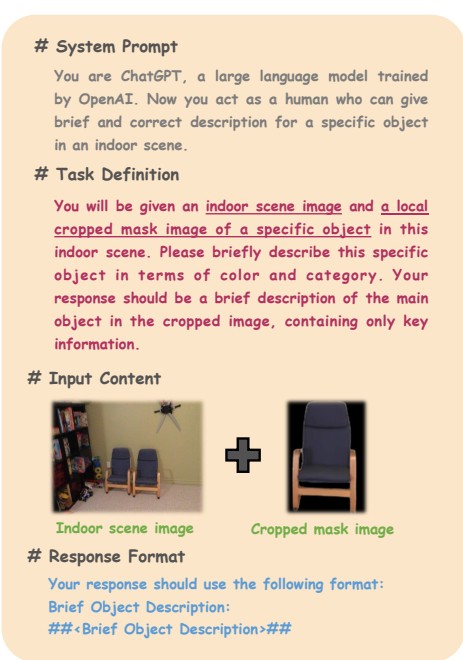

Figure 5: **LLM Prompt.** In this figure, we provide a detailed description of the prompt and various requirements that we input into our LLM agent.

## A    FURTHER IMPLEMENTATION DETAILS

**Statement of the use of LLM.** In our framework, considering the language modality, we employ GPT-4o Hurst et al. (2024) as an LLM agent to analyze observed objects in the 2D query image and automatically annotate language descriptions for them. We elaborate on the prompt content we input into this LLM agent in Fig. 5. In addition to providing both the object itself and its surrounding environment as input, we also restrict the output format of the LLM agent to prevent excessively long responses. CLIP has trouble with processing too long sentences, often resulting in information confusion or semantic drift, which has a negative impact on language descriptors and landmark association. Thus, LLM plays an *'Object-Agent'* module to provide language modality through common-sense reasoning. Meanwhile, LLM is also used for linguistic polish and grammar checks.

**Pose optimization.** During object-level pose optimization, due to the limited number of object correspondences, the presence of outliers is very detrimental to the final result. Therefore, in the optimization process, we also apply a median filter strategy to remove matching pairs with an ICP loss larger than $5\times$ the median loss value. This enables a more stable pose refinement.

## B    OPEN-VOCABULARY OBJECT MATCHING

First, as an object-level method, ObjLoc can decompose diverse objects in a scene. Low-level vision solutions naturally fall short in such capabilities, which reveals their essential difference from ours. Furthermore, as an open-vocabulary method, ObjLoc can recognize class-agnostic objects, involving common or long-tail categories. Oppositely, GoReloc Wang et al. (2024) is restricted to

Table 6: **Object Matching Results.** Quantitative object matching performance on single- and multi-room scenes.

| Dataset (Metric) | Scene1 | Scene2 | Scene3 | Scene4 | Scene5 | Scene6 | Scene7 | Scene8 |
|---|---|---|---|---|---|---|---|---|
| *Synthetic* **(Accuracy[%]↑)** | 88 | 93 | 85 | 89 | 95 | 97 | 89 | 94 |
| Dataset (Metric) | 0568 | 0101 | 0673 | 0108 | 0166 | 0378 | 0092 | 0603 |
| *ScanNet* **(Accuracy[%]↑)** | 85 | 90 | 89 | 89 | 75 | 92 | 91 | 94 |

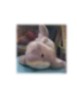 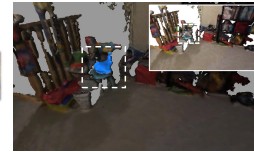 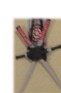 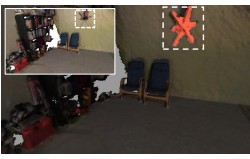 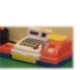 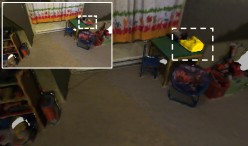


"Gray Toy Dolphin"     "Red and Black Sword Set"     "Multicolored Toy Cash Register"


Figure 6: **Open-Vocabulary Landmark Association.** Open-vocabulary object-level mapping allows our system to leverage rich class-agnostic objects in the landmark association.

Table 7: **Relocalization Results under Varying Lighting.** We compare ours with CoordiNet Moreau et al. (2022) under different lighting conditions. Experiment results illustrate that our method exhibits better lighting invariance, enabling more robust relocalization.

| Setting | Method | @50cm | | | @25cm | | |
|---|---|---|---|---|---|---|---|
| | | Recall[%]↑ | MTE[m]↓ | MRE[°]↓ | Recall[%]↑ | MTE[m]↓ | MRE[°]↓ |
| 50% Lighting | CoordiNet | 40 | 0.34 | 10.5 | 11 | 0.2 | 10.5 |
| | Ours | **72** | **0.13** | **5.3** | **57** | **0.08** | **3.0** |
| 75% Lighting | CoordiNet | 47 | 0.34 | 11.4 | 11 | 0.17 | 11.1 |
| | Ours | **79** | **0.12** | **5.2** | **67** | **0.07** | **2.7** |
| 100% Lighting | CoordiNet | 64 | 0.32 | 6.4 | 13 | 0.15 | 6.5 |
| | Ours | **81** | **0.09** | **3.7** | **70** | **0.06** | **2.0** |

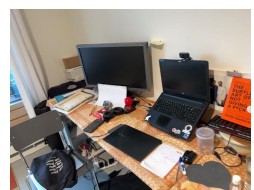 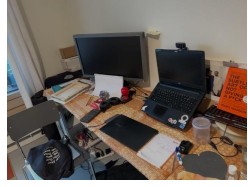 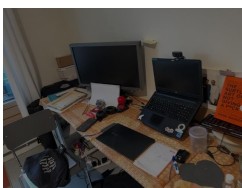


**100% Lighting**     **75% Lighting**     **50% Lighting**


Figure 7: **Lighting Variance.** We respectively display scene appearance under 100%, 75%, and 50% lighting in this figure.

close-vocabulary object sets and can only utilize a small number of categories, such as table and chair. We evaluate our object matching performance by the percentage of successfully matched objects, i.e., $Accuracy[\%]$. This evaluation is performed on open-vocabulary object sets of single- and multi-room scenes (ScanNet and Synthetic) in Tab. 6. We also qualitatively present some matched long-tail objects and their LLM reasoning descriptions in Fig. 6.

## C  LIGHTING VARIANCE ANALYSIS

Object-level camera relocalization is highly robust to appearance changes. Even under varying lighting conditions, our system can still achieve reliable and accurate relocalization based on the geometry structure and semantic information. We evaluate relocalization performance under different lighting conditions in the ScanNet++ Yeshwanth et al. (2023) '8890d' scene. In our experiment, we set a progressive light reduction to 75% and 50%. When the lighting decays significantly, our method only experiences a slight performance drop. In contrast, CoordiNet Moreau et al. (2022) shows a noticeable decline of 20% in success rate. Experiment results in Tab. 7 demonstrate that our system can effectively handle drastic lighting variance, further demonstrating its great potential for long-term applications. We also qualitatively display some observations with reduced lighting in Fig. 7.

Table 8: **Ablation on Invalid Object Filtering.** The experiment results demonstrate the necessity of filtering invalid objects in an object-level camera relocalization system.

| Method | @50cm | | | @25cm | | |
|---|---|---|---|---|---|---|
| | Recall[%]↑ | MTE[m]↓ | MRE[°]↓ | Recall[%]↑ | MTE[m]↓ | MRE[°]↓ |
| w/o Invalid Object Filtering | 77 | **0.18** | 4.2 | 55 | **0.13** | 2.9 |
| w/ Invalid Object Filtering (**Ours**) | **79** | **0.18** | **4.0** | **58** | **0.13** | **2.9** |

Table 9: **Ablation on DIOU-based Retrieval.** The experimental results demonstrate that the DIOU metric is more appropriate for an object-level camera relocalization system.

| Setting | @50cm | | | @25cm | | |
|---|---|---|---|---|---|---|
| | Recall[%]↑ | MTE[m]↓ | MRE[°]↓ | Recall[%]↑ | MTE[m]↓ | MRE[°]↓ |
| Completeness | 74 | 0.19 | 4.2 | 50 | **0.13** | 3.0 |
| DIOU (**Ours**) | **79** | **0.18** | **4.0** | **58** | **0.13** | **2.9** |

Table 10: **Ablation on Relocalization Losses.** The experimental results demonstrate the superiority and rationality of our loss design.

| Setting | @50cm | | | @25cm | | |
|---|---|---|---|---|---|---|
| | Recall[%]↑ | MTE[m]↓ | MRE[°]↓ | Recall[%]↑ | MTE[m]↓ | MRE[°]↓ |
| Center Alignment | 62 | 0.24 | 6.9 | 34 | 0.14 | 4.9 |
| 2D ICP + Center Alignment | 75 | 0.19 | 4.3 | 51 | **0.13** | 3.0 |
| 2D ICP (**Ours**) | **79** | **0.18** | **4.0** | **58** | **0.13** | **2.9** |

## D    FURTHER ABLATION STUDIES

For other designs in our system, such as invalid object filtering, DIOU-based retrieval, and relocalization losses, we investigate their effectiveness on the ScanNet '0568' scene.

**Effectiveness of Invalid Object Filtering.** Invalid objects disrupt our landmark association and global scene graph structure; for example, the wall or floor is connected to almost all nodes in a scene graph. Results in Tab. 8 show that we can improve relocalization performance by saving only objects that are valuable for relocalization. We also visualize filtering results in Fig. 8. Both quantitative and qualitative experiments validate the effectiveness of the invalid object filtering.

**Effectiveness of DIOU-based Retrieval.** To explain the contribution of the DIOU-based retrieval strategy, we set a completeness strategy as a comparison. Specially, the completeness strategy votes for a reference frame based on the straightforward completeness of matched landmarks. As shown in Tab. 9, our proposed DIOU-based retrieval can indeed provide a more appropriate pose prior for the query image, thereby enhancing the object-level tracker.

**Effectiveness of Relocalization Losses.** We assess three different pose estimation losses in Tab. 10, including center alignment loss, 2D ICP loss and their weighted combination. The recall and accuracy achieved by our 2D ICP design are much better than those of the others. The results demonstrate the compatibility of our 2D ICP loss with the object-level camera relocalization task, as well as the negative effect caused by center alignment loss.

## E    FURTHER SYSTEM ANALYSIS

In this section, we further compare our method with PixLoc Sarlin et al. (2021), a SOTA approach that estimates camera poses from low-level feature point correspondences. Unlike our system, PixLoc requires substantially more memory and lacks high-level scene understanding, which makes direct comparison less informative. We nevertheless include this feature-based baseline to provide a more comprehensive analysis. While ObjLoc emphasizes semantic and geometric cues, PixLoc primarily relies on appearance information. To ensure equal input quality in appearance and geometry, we choose the Synthetic dataset for evaluation and report average results across all scenes in Tab. 11.

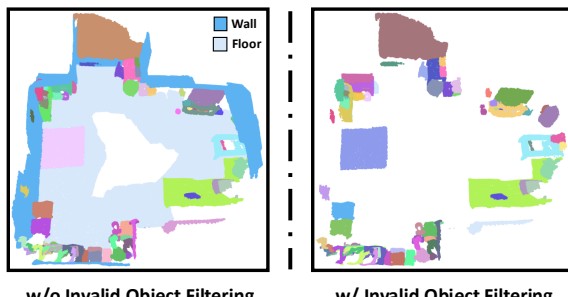

Figure 8: **Invalid Object Filtering.** In this figure, it can be observed that our invalid object filtering module has correctly and cleanly removed the invalid floor and wall.

Table 11: **Comparison with PixLoc on the Synthetic Dataset.** This table shows average relocalization results and map size across all multi-room/floor scenes.

| Method | @50cm | | | @25cm | | | *Map Size* | *Single-frame Runtime* |
|---|---|---|---|---|---|---|---|---|
| | Recall[%]↑ | MTE[m]↓ | MRE[°]↓ | Recall[%]↑ | MTE[m]↓ | MRE[°]↓ | [MB]↓ | [s]↓ |
| PixLoc | 82 | **0.03** | **1.3** | 80 | **0.02** | **1.0** | 273.8 | ≈**4.5s** |
| **Ours** | **87** | 0.10 | 3.9 | **81** | 0.06 | 2.2 | **24.5** | ≈6.0s |

In terms of success rate, $Recall@50cm$ and $Recall@25cm$ indicate that ObjLoc can successfully recall more query frames, as semantic object matching helps mitigate the appearance sensitivity. In terms of accuracy, since object-level correspondences are inherently less fine-grained than point-level ones, ObjLoc exhibits slightly lower accuracy ($MTE$ and $MRE$). In terms of map size, PixLoc Sarlin et al. (2021) requires a large number of RGB reference frames for pose priors and a pre-trained CNN for point-wise low-level features, whose memory consumption exceeds ours by approximately **1000%**. In terms of runtime, as a training-free method, our system achieves comparable efficiency to PixLoc. It is worth noting that LLM reasoning consumes a considerable portion of inference time (**2∼3s**) in our system. This is because GPT-4o is used through an API call in our system, which incurs a request latency. Our system still holds efficiency potential through some engineering tricks, such as CUDA acceleration, half-precision calculation, and local deployment of the LLM agent.

In summary, for the camera relocalization task, ObjLoc can robustly recall camera poses based on scene objects and provide a compact and semantically rich map representation, thereby presenting an essentially different technical route compared to low-level vision methods.

## F    MORE VISUALIZATION RESULTS

**Relocalization Results.** Similar to Fig. 4 in the main paper, we visualize more relocalization results obtained by our system across all datasets in Fig. 9. It is evident that our method can recover camera poses that closely align with the ground truth in different indoor regions, which demonstrates the superior accuracy and robustness of our system.

**Instance Segmentation Results.** The instance segmentation module initially identifies all objects within a scene, playing a crucial role in determining the quality of object-oriented mapping. As illustrated in Fig. 10, we qualitatively present the instance segmentation performance of our system on different scenes. These visualization results demonstrate that our system is able to perform thorough and detailed segmentation for diverse objects with varying sizes.

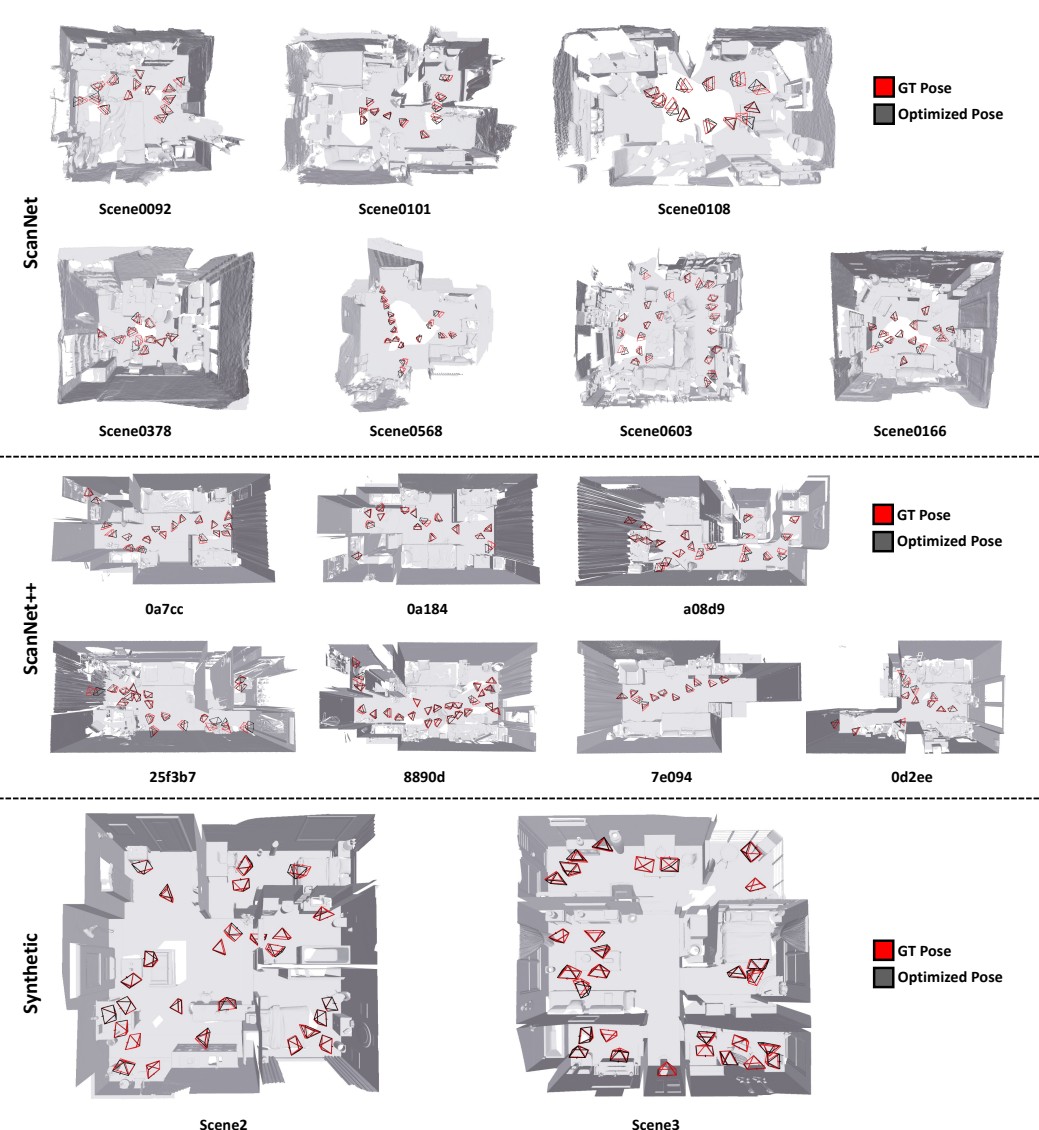

Figure 9: **Relocalization Visualization.** We qualitatively show our relocalization results on various datasets in this figure.

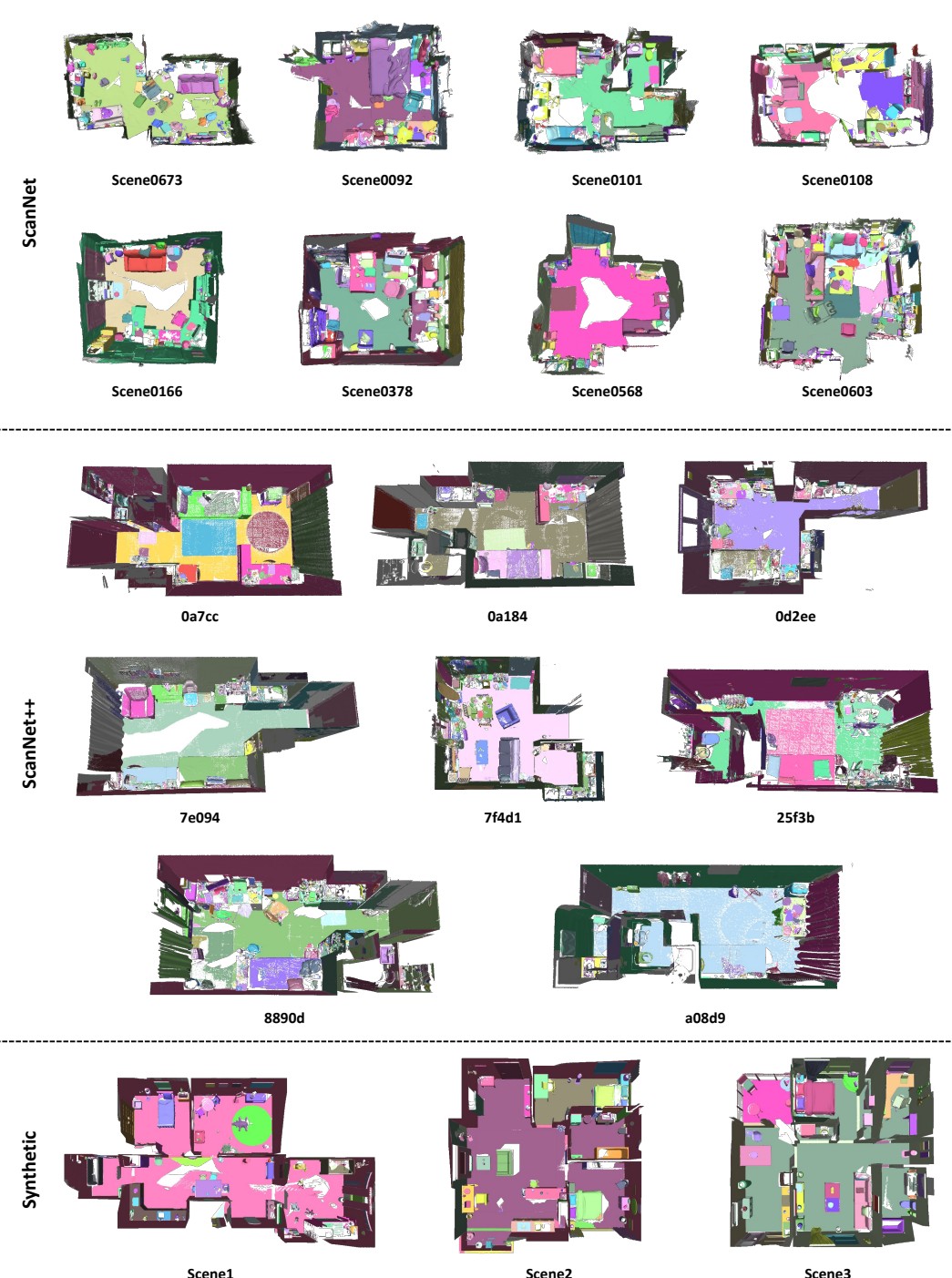

Figure 10: **Instance Segmentation Visualization.** We qualitatively show our instance segmentation results on various datasets in this figure.