# OpenReview forum: "ObjLoc: Indoor Camera Relocalization based on Open-Vocabulary Object-Level Mapping"
_ICLR.cc/2026/Conference — ICLR 2026 Conference Withdrawn Submission_

### Official Review · Reviewer_nktE · 2025-10-27

**Soundness:** 3
**Presentation:** 3
**Contribution:** 3
**Rating:** 6
**Confidence:** 4

**Summary:**

ObjLoc is an open-vocabulary, object-level indoor camera relocalization system that integrates semantic object understanding, scene graph reasoning, and 2D–3D pose optimization to achieve scalable and interpretable localization. The system leverages open-vocabulary models (Florence2, CLIP) to detect and encode objects, and enhances semantic comprehension through text descriptions generated by GPT-4o. Meanwhile, it constructs a scene graph to model spatial relationships between objects. Localization is performed in a coarse-to-fine manner, where a DIoU-based strategy and a newly proposed dual-path 2D ICP loss are employed to align observed and projected object pixel regions, enabling accurate pose estimation.

**Strengths:**

Originality: ObjLoc proposes an open-vocabulary, object-level camera relocalization framework that integrates semantic understanding, geometric consistency, and scalable map construction at the system level. The introduced object-level map suite and dual-path 2D ICP loss represent a clear departure from traditional keypoint- or patch-based localization paradigms, demonstrating strong originality.

Quality: The paper replaces full images with object-based reference frames, extracts features using CLIP, applies a DIoU-based strategy for coarse pose filtering, and employs a dual-path ICP for fine-grained pose optimization. These components are rigorously validated through comprehensive experiments, showing strong robustness and reliability.

Clarity: The paper is well-structured with clear motivation and a modular system design. The entire pipeline—from map construction and object association to pose refinement—is logically organized and easy to follow. Illustrations and qualitative results visually demonstrate the workflow and innovations, making the presentation clear and coherent.

Significance: The paper effectively addresses key challenges in object-level camera relocalization, such as low descriptor discriminability, lack of reliable pose priors, and ambiguity from center-point alignment. Its proposed solutions achieve superior performance over existing methods, offering both strong research significance and practical application potential.

**Weaknesses:**

１.While the system demonstrates strong performance in terms of accuracy and scalability, the paper does not discuss computational efficiency in the experiments. The inference time of the open-vocabulary, object-level relocalization method is not reported.

２.The paper employs object-level feature descriptors based on CLIP, but it does not compare against traditional feature-based or dense matching methods, which raises concerns about the credibility of the reported performance.

３.The paper proposes using object-level reference frames as a replacement for RGB images to reduce memory consumption. However, it does not provide a comparative experiment to demonstrate whether this substitution leads to improved performance.

**Questions:**

１.The system seems to rely heavily on the detection and encoding of objects via CLIP. How does ObjLoc perform when applied to scenes with a diverse set of object types, especially in environments with uncommon or rare objects that might not be well-represented in the model's training data?

２.Compared with dense feature-based methods, does the object-level representation limit the localization accuracy?

３.How does ObjLoc perform across different indoor benchmarks, such as 7Scenes? A comparison with existing state-of-the-art methods on these datasets would help validate the claims of superior performance.

４.Can ObjLoc be extended to outdoor or larger-scale environments? Have any related tests or improvements been conducted?

---

### Official Review · Reviewer_JSej · 2025-10-31

**Soundness:** 3
**Presentation:** 2
**Contribution:** 2
**Rating:** 2
**Confidence:** 4

**Summary:**

The proposed method builds a system to query the camera pose of an RGB image with a given 3D segmentation map. Given a sequence of posed RGB-D images, the author first clusters the 2D segmentation into 3D space to form a scene graph containing point cloud, bounding box and clip feature for each object. During camera relocalization, the Florence-2 and GPT-4o will be used to extract the visual cue and language cue from the query image. Then the camera will be relocalized by sub-graph matching between the query image and the 3D scene graph. The final pose is further refined by ICP.

**Strengths:**

The results look good.

**Weaknesses:**

The paper is hard to follow. Furthermore, the novelty of the method is limited. It seems more suitable for a robotics conference like IROS and ICRA rather than ICLR. The author uses SAM, CLIP and Cropform to build the scene graph, while during relocalization, the author only uses Florence-2 and GPT-4o to extract features from the query image. Why not use SAM, CLIP and Cropform as well?
Comparisons with some baselines are missing, like DSAC, DSAC++, ACE, ACE0.

More detailed questions:
Line 195, the description of the strategy of mask nodes is missing. The author should clarify whether it is the same as mask clustering.
Fig 3, The explanation of subgraph similarity calculation is missing. Furthermore, the term used in this section is ambiguous. If the subgraph similarity is calculated as Fig 3(a), it cannot be represented as "compute the similarity of all possible neighbor pairs". Especially in line 266, the author claims the subgraph has a length n. Also, the object candidate of $G_q$ is not defined, which indicates Fig 3(a) might be wrong. If  $G_q$  is the graph on the query image, the objects in Fig 3(a) would be outside the query images.
Sec 3.3, the coarse pose prior is derived from the pose of the reference frame with respect to the 2D box, which is sensitive to outliers.
Equ 7b, the definition of $p_i$ is missing. The operation of ψis missing.
In experiments, the details of reference frame selection are missing.
The iteration number/optimizer of 2D ICP is missing.
The visualization of segmentation, Fig 10, is irrelevant to the proposed method.

**Questions:**

Due the to limitations mentioned in the weaknesses, I tend to give negative recommendation.

---

### Official Review · Reviewer_KRu6 · 2025-11-01

**Soundness:** 3
**Presentation:** 3
**Contribution:** 3
**Rating:** 6
**Confidence:** 3

**Summary:**

This paper introduces ObjLoc, an object-level indoor camera relocalization framework that builds a compact, semantic 3D map using open-vocabulary object descriptors (CLIP), scene graphs, and object reference frames. At inference, ObjLoc performs vision–language matching for 2D–3D correspondence, retrieves a coarse pose via DIOU-based matching (to the reference frames), and refines it using a dual-path 2D ICP loss. Experiments on multiple indoor datasets demonstrate significant improvements in accuracy and recall while using smaller maps.

**Strengths:**

1. Overall, the paper is well-written and clearly structured.

2. Integrates vision–language matching, scene-graph context, and geometric optimization into a unified pipeline. The proposed dual-path 2D ICP loss and DIOU-based retrieval are elegant and well-motivated.

3. Demonstrates strong robustness and scalability across multi-room and multi-floor indoor scenes.

4. The object-oriented reference frame design leads to highly compact and storage-efficient maps.

5. The authors include thorough ablation studies that help clarify the impact of key design choices.

**Weaknesses:**

1. The use of multiple large models (CLIP + SAM + Florence2 + GPT-4o) results in slow runtime (6.0 seconds per frame) and large memory requirements, making real-time operation unrealistic.

2. The authors did not compare their method to strong modern visual localization systems such as ACE or GLACE. Given their performance and relevance, such a comparison is necessary to provide context for the contribution.

3. Since the method relies exclusively on object-level cues for localization, it is unlikely to perform well in dynamic environments (as briefly acknowledged in the limitations). This severely limits its practicality.

4. The pipeline depends heavily on accurate instance segmentation and object detection; any segmentation or detection error can propagate through the entire system, leading to incorrect associations or failed relocalization. A robustness analysis against such errors is missing.

- ACE: Learning to Relocalize in Minutes using RGB and Poses, CVPR 2023.
- GLACE: Global Local Accelerated Coordinate Encoding, CVPR 2024.

**Questions:**

1. I would highly appreciate a comparison to more modern and stronger visual localization baselines such as ACE and GLACE, as well as the addition of median rotation and translation errors as evaluation metrics (computed over all samples, not only those under the threshold).

---

### Official Review · Reviewer_5fUS · 2025-11-04

**Soundness:** 2
**Presentation:** 3
**Contribution:** 2
**Rating:** 0
**Confidence:** 5

**Summary:**

The paper presents a visual localization pipeline using an object-level map. The mapping stage using RGB-D images creates a graph map, where each node is an open-vocabulary object defined by its 3D point cloud, corresponding keyframes, and CLIP-based descriptor. The localization stage first establishes correspondences between objects in the query and in the map, which are then used to estimate the query camera pose in a coarse-to-fine manner - starting from the pose of the best-matching keyframe and optimizing the object projection alignment using ICP. The main contribution of the paper is the overall design of the pipeline and a number of partial design choices (descriptor, retrieval, ICP), which might be interesting for the design of future object-level localization pipelines.

**Strengths:**

The paper tackles an interesting problem. In particular, I appreciate the idea of building abstract (in terms of using semantics / segmentations rather than appearance information) and thus compact scene representations.

The paper builds on ideas presented in previous works (GOReloc, CLIP-Loc), but the overall design is interesting and novel in terms of using language information and DIOU-based retrieval. The ideas presented in the paper (combined language and image-based CLIP descriptor, DIOU retrieval, and ICP alignment based on object point clouds) could be interesting to the community and for future works on object-level camera localization.

The proposed approach is technically sound. The paper is well-written and easy to follow. The presented figures help understand the design.

**Weaknesses:**

**Baselines selection and SotA claims.**

The paper claims that "Experiments on various datasets demonstrate that our object-level system consistently achieves state-of-the-art performance in indoor camera relocalization" (L107). Yet, this is not true for multiple reasons:

1. The paper does not contain a comparison to any current state-of-the-art localization methods. In fact, it does not even cite the two probably gold-standard methods for accurate visual localization: hloc [Sarlin et al., From Coarse to Fine: Robust Hierarchical Localization at Large Scale, CVPR 2019] [Sarlin et al., SuperGlue: Learning Feature Matching with Graph Neural Networks, CVPR 2020] (a method based on matching local features) and ACE [Brachmann et al., Accelerated Coordinate Encoding: Learning to Relocalize in Minutes Using RGB and Poses, CVPR 2023] (a learning-based scene coordinate regressor). ACE (and other similar scene coordinate regressors) easily achieve similar map sizes as the proposed method. hloc by default produces a large map representation, but recent work [Wang, MAD-DR: Map Compression for Visual Localization with Matchness Aware Descriptor Dimension Reduction, ECCV 2024] [Laskar et al., Differentiable Product Quantization for Memory Efficient Camera Relocalization, ECCV 2024] shows that these representations can be reduced to a few MB without too large a loss in accuracy. Both hloc and ACE have publicly available source code.

Two of the baselines (Moreu et al., and Shavit et al.) belong to the family of absolute camera pose regressors. It is well-known by now that approaches from this family of localization algorithms perform significantly worse than methods based on 3D models. In fact, it has been shown that such approaches might not even outperform simple pose approximation strategies (see [Sattler et al., Understanding the Limitations of CNN-based Absolute Camera Pose Regression, CVPR 2019]).

2. The paper evaluates the proposed approach on datasets that are not commonly used in the localization literature, making it virtually impossible to compare its performance with state-of-the-art approaches. It is unclear why more commonly used datasets such as 7Scenes, 12Scenes, and RIO10 (all room-level), Indoor-6 (multiple rooms) InLoc (two floors of a large university building), or the Gangnam Station and Hyundai Department Store datasets (both multiple floors of larger indoor scenes) were not used. Compared to ScanNet and ScanNet++, they offer the advantage that mapping and query images were obtained from different trajectories, making the localization problem harder. RIO10, InLoc, and Indoor-6 further offer query images taken under different viewing and illumination conditions, again making the visual localization problem harder.

3. There are multiple works based on refining an initial pose estimate. The supplementary material compares to PixLoc, which was one of the first such methods based on learned features. In particular, the work by Pietrantoni et al. seems to be highly relevant to this work under review, In the context of privacy-preserving visual localization, they aim to estimate the pose of a query image wrt. an abstract map. The map they are using is represented by a set of segmentation labels and corresponding 3D points (e.g., from sparse structure-from-motion points in [Pietrantoni et al., Segloc: Learning segmentation-based representations for privacy-preserving visual localization, CVPR 2023] or dense Gaussian Splats in [Pietrantoni et al., Gaussian Splatting Feature Fields for (Privacy-Preserving) Visual Localization, CVPR 2025]). After obtaining an initial pose estimate via image retrieval, they refine the pose estimate by iteratively updating the camera pose such that the segment labels detected in the query image and a projection of the segment labels stored in the map match as good as possible (a similar cost function was used in [Lianos et al., VSO: Visual Semantic Odometry, ECCV 2018] in the context of visual SLAM). Since they do not need to store colors or features, these methods produce compact representations.

In general, the idea of localizing wrt. a semantic map (storing semantic labels together with 3D geometry) was already explored in [Stenborg et al., Long-term visual localization using semantically segmented images, ICRA 2018], [Toft et al., Semantic Match Consistency for Long-Term Visual Localization, ECCV 2018], and [Larsson et al., A Cross-Season Correspondence Dataset for Robust Semantic Segmentation, CVPR 2019].

These methods are all highly relevant, not cited in the work under submission, and would make natural baselines for a family of methods highly related to the proposed approach.

In summary, the current experimental evaluation is clearly insufficient for understanding the performance of the proposed approach compared to related methods and the current state-of-the-art.

**Assumption on static scene.**

The paper mention that the method "cannot capture and locally update dynamic changes in a scene" (L484). It is not clear if this applies only to the mapping phase or even if the localization does not work in the presence of scene changes. If this is the second case, the limitation is very significant, and it should be mentioned prominently right at the beginning of the paper. In particular, most of the approaches discussed above are robust against such changes at query time.

**Lighting Variance Analysis.**

Sec. C of the supplementary claims to test localization under different lighting conditions. However, the example in Fig. 7 shows three images that seem to have adjusted brightness, not changed lighting in the scene. The authors should clarify if they approximate the lighting changes by adjusting image brightness. If this is the case, using existing dataset that contains real-world changes would be preferable (see list of datasets above).

**Questions:**

In its current form, without comparisons to state-of-the-art methods and discussions and comparisons to highly related methods, I do not see how the paper can be accepted. In order to change my recommendation, I would want to see such comparisons (at least with hloc and ACE) on established datasets (the Indoor-6 dataset (https://github.com/microsoft/SceneLandmarkLocalization) seems small enough to allow evaluation in limited time but is challenging enough that results would be meaningful (as opposed to the 7Scenes and 12Scenes datasets which are essentially solved)).

---

### Note · Authors · 2025-11-12

I have read and agree with the venue's withdrawal policy on behalf of myself and my co-authors.